# T-measure: A Topology-Consistent Metric for Binary Segmentation

**Pengfei Zhang** [1]    **Jian Ji** [1]

## Abstract

Evaluation metrics establish a standard assessment framework for models, playing a pivotal role in model optimization and advancement. However, widely adopted pixel-wise metrics (e.g., IoU, Dice) rely heavily on pixel-level statistics, often failing to capture the structural integrity of predictions. While the S-measure ($S_m$) incorporates structural perception to some extent, it struggles to differentiate critical structural violations and remains insensitive to background false positives and small objects. To address these limitations, we propose the Topology-aware measure ($T_m$), a novel metric designed to explicitly quantify topological consistency. $T_m$ employs the Fuzzy Jaccard Index as a foundational score, integrates a Topological Integrity term ($\mathcal{I}_{topo}$) to penalize critical structural fragmentation, and utilizes a Boundary Alignment term ($\mathcal{A}_{bdy}$) to evaluate boundary alignment. These three components synergize to achieve robust evaluation of prediction maps at the topological level. We establish a rigorous Meta-Measure validation framework and benchmark our method against nine mainstream metrics across diverse complex scenarios. Extensive experiments demonstrate that $T_m$ performs exceptionally in downstream tasks and maintains high consistency with human visual perception. The code and evaluation toolkit are available at https://github.com/Higte/Tm.

## 1. Introduction

Binary image segmentation serves as a cornerstone in computer vision, with applications spanning multiple domains such as Salient Object Detection(Qin et al., 2020), Camouflaged Object Detection(Fan et al., 2022), Industrial Defect Detection(Bergmann et al., 2019), and Medical Image Segmentation(Ronneberger et al., 2015). In recent years, image segmentation techniques have advanced rapidly. While modern models are capable of perceiving contextual relationships within images and capturing high-level semantic information of objects to complete segmentation tasks. Unfortunately, currently widely used evaluation metrics remain limited to pixel-level counting. They treat prediction results as unstructured collections of pixels, thereby ignoring the structural fidelity and geometric completeness achieved by state-of-the-art models.

For example, widely adopted metrics such as Intersection over Union(Jaccard, 1901)($IoU$), Dice coefficient(Dice, 1945)($Dice$), and Mean Absolute Error(Perazzi et al., 2012)($MAE$), while effective in measuring region-based statistics, treat pixels as independent entities, thereby neglecting the structural integrity of segmented objects. As shown in the first row of Figure 1, $IoU$ assigns comparable scores to a high-quality prediction and a completely fragmented one. Even metrics that incorporate global statistics, such as E-measure(Fan et al., 2018)($E_m$), fail to effectively validate structural validity. As illustrated in the second row of Figure 1, as long as the pixel count remains similar, $E_m$ assigns a high score of 0.891 even if the prediction map is merely meaningless random noise. This result fundamentally contradicts human visual perception.

To mitigate the limitations of pixel-level evaluation, researchers have proposed several structure-aware metrics. In the field of medical image segmentation, Hausdorff Distance(Huttenlocher et al., 1993)($95HD$) and Average Surface Distance(Taha & Hanbury, 2015)($ASD$) are commonly used. Although they reflect certain boundary structural information, their extreme sensitivity to outliers results in a lack of stability in general evaluation. In the field of Salient Object Detection, S-measure ($S_m$) is frequently used to evaluate the structural similarity of segmentation results. However, it struggles to distinguish between critical structural violations and non-critical geometric errors. Furthermore, its calculation mechanism renders $S_m$ insensitive to background ghosting. As shown in the third row of Figure 1, $S_m$ still yields scores comparable to those of high-quality prediction maps when facing obvious background errors.

Based on the aforementioned issues, we propose the

[1]School of Computer Science and Technology, Xidian University, Xi'an, Shaanxi, China. Correspondence to: Jian Ji <jji@xidian.edu.cn>.

*Proceedings of the 43rd International Conference on Machine Learning*, Seoul, South Korea. PMLR 306, 2026. Copyright 2026 by the author(s).

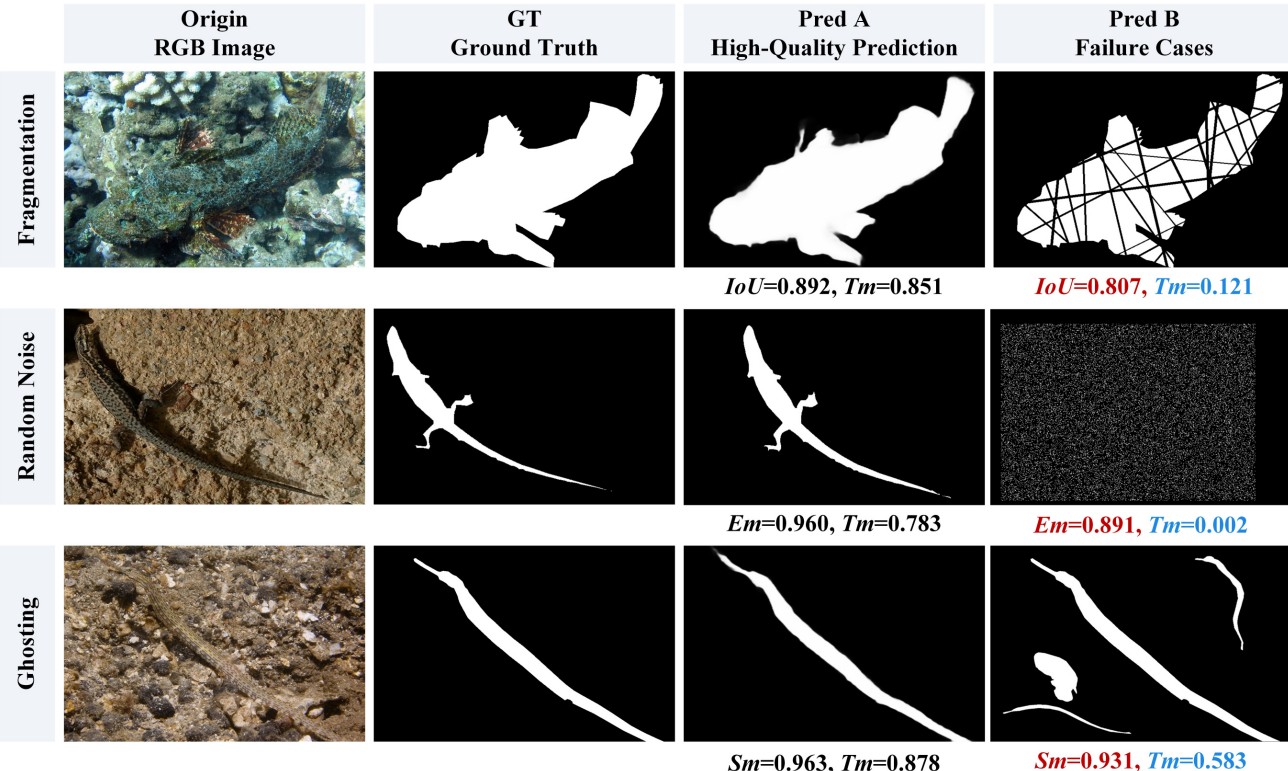

*Figure 1*. Visual comparison of $T_m$ with existing metrics across three failure modes. Row 1: $IoU$ assigns comparable scores to a structurally intact prediction and a severely fragmented one, failing to penalize critical structural damage. Row 2: For a meaningless random noise map generated using the same pixel count as the high-quality prediction, $E_m$ yields a score of 0.891. Row 3: $S_m$ is insensitive to background ghosting, failing to effectively penalize obvious errors. In all scenarios, $T_m$ imposes effective penalties.

Topology-aware measure ($T_m$), a novel metric designed to explicitly quantify both topological fidelity and geometric precision. $T_m$ is formulated as a unified framework integrating three synergistic components: (1) a Fuzzy Jaccard Index ($S_{pix}$) as a foundational term for robust soft-evaluation of pixel probability maps; (2) a Topological Integrity term ($\mathcal{I}_{topo}$) that utilizes connected component analysis to penalize topological fragmentation; and (3) a Boundary Alignment term ($\mathcal{A}_{bdy}$) designed to adaptively evaluate boundary conformity. Leveraging these core components, $T_m$ evaluates prediction quality with a specific emphasis on topological integrity, a pivotal yet long-overlooked dimension in binary segmentation assessment. To validate the effectiveness of $T_m$, we established a rigorous Meta-Measure validation framework to conduct comprehensive testing across diverse domains, objects, and scenarios.

Our main contributions are summarized as follows:

(1) We analyze representative cases of existing metrics in practical applications and identify their deficiencies in the dimension of topological awareness.

(2) We propose the Topology-aware measure ($T_m$), the first metric that explicitly incorporates topological integrity constraints into the scoring mechanism.

(3) We establish a Meta-Measure validation framework to benchmark $T_m$ against nine metrics across four binary segmentation domains. Experimental results demonstrate that $T_m$ exhibits significant advantages in structural awareness and maintains high consistency with human perception.

## 2. Current Evaluation Measures

To provide a comprehensive review of the existing evaluation landscape, we selected nine mainstream metrics from four domains: Salient Object Detection, Camouflaged Object Detection, Industrial Defect Detection, and Medical Image Segmentation. These metrics can be broadly categorized into three types: pixel-wise metrics, enhanced pixel-wise metrics, and structure-aware metrics.

### 2.1. Pixel-wise Metrics.

Pixel-wise metrics constitute the most fundamental and widely adopted measures, including $IoU$, $MAE$, Dice Coefficient($Dice$), and F-measure(Achanta et al., 2009)($F_m$). These metrics typically treat pixels as independent and identically distributed (i.i.d.) samples, calculating scores based on the confusion matrix or pixel-wise absolute

differences. For instance, $IoU$ is defined as:

$$IoU = \frac{TP}{TP + FP + FN},\qquad(1)$$

where $TP$, $FP$, and $FN$ denote standard confusion matrix components. While effective at quantifying regional overlap and pixel-level accuracy, these methods inherently neglect the spatial correlation between pixels. By treating pixels as isolated statistical elements, these metrics lack a holistic perspective, thereby failing to evaluate structural integrity.

## 2.2. Enhanced Pixel-wise Metrics.

To mitigate the limitations of the independence assumption, several metrics introduce weighting mechanisms to incorporate spatial context or global statistics. $F_\beta^w$ (Margolin et al., 2014) integrates pixel dependency and position importance. This metric imposes higher penalties on erroneous predictions that are spatially distant from the target regions, thereby enhancing its ability to characterize spatial structural information to some extent.

$$F_\beta^\omega = \frac{(1 + \beta^2)Precision^\omega \cdot Recall^\omega}{\beta^2 \cdot Precision^\omega + Recall^\omega},\qquad(2)$$

where $\beta$ balances precision and recall, the superscript $\omega$ denotes a spatial weight assigned to each pixel based on Euclidean distance.

Similarly, $E_m$ proposes an enhanced alignment matrix that accounts for both pixel-level matching and image-level statistics. By subtracting the global mean, $E_m$ attempts to capture global binary distribution differences alongside local precision:

$$E_m = \frac{1}{W \times H} \sum \sum \phi_{FM}(x, y),\qquad(3)$$

where $\phi_{FM}$ represents the pixel-wise alignment score derived from bias matrices. Despite these enhancements, $F_\beta^w$ relies primarily on distance-based weights, while $E_m$ relies on mean-based alignment. Consequently, they remain grounded in geometric proximity or intensity statistics, lacking explicit perception of high-level topological information.

## 2.3. Structure-aware Metrics.

Recognizing the importance of structural fidelity, recent works have proposed metrics targeting object structure. In medical segmentation, boundary-based metrics such as $95HD$ and $ASD$ are prevalent. These metrics quantify the Euclidean distance between predicted and ground-truth contours. While reflecting boundary quality, their reliance on distance makes them highly unstable in the presence of outliers, and their disregard for regional overlap limits their general applicability.

$S_m$ stands as the prominent structure-oriented metric in Salient Object Detection and Camouflaged Object Detection tasks. It combines region-aware structural similarity ($S_r$) and object-aware structural similarity ($S_o$):

$$S_m = \alpha \cdot S_o + (1 - \alpha) \cdot S_r.\qquad(4)$$

Specifically, $S_r$ employs SSIM-like comparisons to capture texture layout, while $S_o$ assesses global distribution using probability statistics. It is important to note that the "structure" perceived by $S_m$ is fundamentally a statistical structure, rather than a topological one. Moreover, due to its excessive reliance on area weighting, $S_m$ exhibits significant deficiencies in handling false positive errors and small objects. We provide a detailed analysis of $S_m$ in Section 3.2.

# 3. Limitations of Existing Metrics

## 3.1. Insensitivity to Topological Integrity

To investigate the sensitivity of existing metrics to topological integrity, we designed a controlled experiment as shown in Figure 2. We generated two predictions, P1 and P2, for a Ground Truth (GT) possessing semantic structure. P1 simulates a simple boundary loss, while P2 simulates a critical topological fracture.

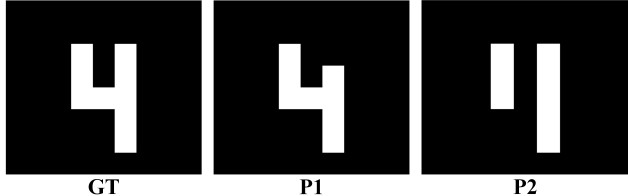

**GT**       **P1**       **P2**

*Figure 2.* Illustration of test samples. GT: Ground Truth (the semantic shape of the digit "4"). P1: A prediction with partial regional loss that maintains the overall topological structure. P2: A prediction exhibiting a critical topological fracture. Note that P1 and P2 have identical missing pixel areas.

*Table 1.* Quantitative comparison of evaluation results of various metrics on the predictions in Figure 2. The arrows (↑ / ↓) indicate whether higher or lower scores are preferable. The $\Delta\%$ column denotes the relative score change from P1 to P2. Note: Unless otherwise specified, all references to $E_m$ and $F_m$ in this paper denote their mean values. Metrics requiring binarization use a standard threshold of 0.5.

| Metric | P1 | P2 | $\Delta\%$ |
|---|---|---|---|
| $MAE \downarrow$ | 0.013 | 0.013 | 0.0% |
| $IoU \uparrow$ | 0.889 | 0.889 | 0.0% |
| $Dice \uparrow$ | 0.941 | 0.941 | 0.0% |
| $F_m \uparrow$ | 0.969 | 0.969 | 0.0% |
| $E_m \uparrow$ | 0.979 | 0.979 | 0.0% |
| $F_\beta^w \uparrow$ | 0.942 | 0.943 | +0.1% |
| $S_m \uparrow$ | 0.939 | 0.947 | +0.9% |
| $95HD \downarrow$ | 27.00 | 12.65 | -53.1% |
| $ASD \downarrow$ | 2.93 | 1.26 | -57.0% |
| $T_m \uparrow$ | **0.872** | **0.607** | **-30.4%** |

To ensure a controlled comparison, P1 and P2 possess identical pixel-level error areas. The fundamental distinction lies in the fact that P1 preserves the majority of the object structure, whereas P2 severely compromises the semantic structural information. Consequently, P2 should be assigned a lower evaluation score. Table 1 reports the evaluation results.

First, the five metrics $IoU$, $MAE$, $Dice$, $F_m$, and $E_m$ assign identical scores to P1 and P2. This confirms that these metrics treat pixels as independent statistical elements. Lacking a structural perspective, they are incapable of distinguishing differences in the topological dimension.

Second, $F_\beta^w$ counter-intuitively assigns a higher score to P2. This is attributed to the fact that $F_\beta^w$ incorporates neighborhood dependency, typically assigning higher importance to boundary regions than to internal smooth regions. Consequently, the boundary loss in P1 is deemed more severe than the structural fracture in P2.

Finally, among the three structure-aware metrics, both $95HD$ and $ASD$ failed in this evaluation, anomalously assigning higher scores to P2. This indicates that defining error solely via Euclidean distance between boundaries is ineffective in capturing high-level structural information. $S_m$ assigned highly comparable scores to P1 and P2, indicating its inability to evaluate structural integrity.

### 3.2. Analysis of Limitations of S-measure

$S_m$ is regarded as the state-of-the-art structure-aware metric, yet it possesses three critical limitations.

**Inability to evaluate topological integrity.** As shown in Table 1, $S_m$ assigns a higher score to P2, which contains structural fracture errors. This is because the perception of structure in $S_m$ fundamentally stems from statistical induction at the pixel level. Consequently, $S_m$ cannot distinguish between general boundary errors and critical structural violations, failing to effectively evaluate the structural fidelity of the target.

**Insensitivity to background ghosting.** During calculation, $S_m$ partitions the image into regions, where the scoring within each region relies heavily on foreground area weighting. This directly causes errors in the background to be often diluted or ignored. As shown in the second row of Figure 1, although the model predicts three distinct erroneous objects in the background, the $S_m$ score remains largely unaffected, indicating its inability to effectively penalize false positives.

**Inapplicability to small objects.** The object-aware component ($S_o$) of $S_m$ evaluates the similarity of the foreground and background separately and combines them using foreground area weighting. In scenarios involving small objects, the background occupies the vast majority of the image.

Therefore, even if the model makes no prediction (i.e., outputs a completely black image), $S_o$ can achieve a near-perfect evaluation result solely by correctly classifying the background, ultimately leading to an $S_m$ score approaching 0.5. This violates the visual intuition that "a total miss deserves a zero score," rendering $S_m$ unreliable in small object detection.

## 4. Our Measure

In this section, we present the proposed Topology-aware measure ($T_m$). Formulation. Given a predicted probability map $P \in [0,1]^{H \times W}$ and the ground truth binary mask $G \in \{0,1\}^{H \times W}$, $T_m$ is formulated as a structure-constrained pixel fidelity score:

$$T_m(P, G) = \mathcal{S}_{pix} \times [\alpha \mathcal{I}_{topo} + (1 - \alpha)\mathcal{A}_{bdy}]. \quad (5)$$

For brevity, we omit the arguments $(P, G)$ for the component terms. Here, $\mathcal{S}_{pix}$ represents the base pixel-level similarity derived from fuzzy set theory. The term within the brackets functions as a topological penalty coefficient, composed of the Topological Integrity ($\mathcal{I}_{topo}$) and Boundary Alignment ($\mathcal{A}_{bdy}$). $\alpha \in [0,1]$ is a hyper-parameter that balances the sensitivity between topological connectivity and geometric boundary precision. It is worth noting that binarization is required exclusively for calculating $\mathcal{I}_{topo}$, whereas $\mathcal{S}_{pix}$ and $\mathcal{A}_{bdy}$ can be computed directly from the probability map. Consequently, our metric enables the effective evaluation of raw model outputs.

In practice, we set $\alpha = 0.8$. For a detailed analysis justifying this parameter selection, please refer to Appendix A. Furthermore, $T_m$ adopts a decoupled design, which endows it with error diagnostic capabilities beyond standard evaluation. This enables error attribution analysis of models to a certain extent. Please refer to Appendix C for a more detailed discussion.

### 4.1. Pixel-level Similarity

To ensure the metric's applicability across diverse scenarios and provide a robust baseline, we employ $\mathcal{S}_{pix}$ to measure the global overlap between distributions. Viewing the continuous prediction $P$ and binary ground truth $G$ as fuzzy sets, we adopt the fuzzy intersection-over-union formulation(Csurka et al., 2013). Let $p_i$ and $g_i$ denote the probability values of the $i$-th pixel in $P$ and $G$, respectively. $\mathcal{S}_{pix}$ is calculated as:

$$\mathcal{S}_{pix}(P, G) = \frac{\sum_i (p_i \cdot g_i)}{\sum_i (p_i + g_i - p_i \cdot g_i) + \epsilon}, \quad (6)$$

where we employ the algebraic product ($p_i \cdot g_i$) as the fuzzy intersection (t-norm), and the probabilistic sum ($p_i + g_i - p_i \cdot g_i$) as the fuzzy union (s-norm), where $\epsilon$ is a small constant

included to ensure numerical stability. This term effectively quantifies the spatial overlap, penalizing significant region shifts or omissions.

### 4.2. Topological Integrity

$\mathcal{I}_{topo}$ serves as the critical component of $T_m$, explicitly penalizing topological fractures. We model this by evaluating the consistency of connected components between the prediction and the ground truth.

First, we decompose the ground truth $G$ into a set of $N$ disjoint connected components $\{C_k^G\}_{k=1}^N$, where each $C_k^G$ represents a distinct object instance or region. Simultaneously, we derive a binary proposal map $B_p$ from $P$ to facilitate connected component statistics, using a fixed threshold of $\tau = 0.5$. For each ground truth component $C_k^G$, we examine its corresponding region within the prediction, defined as the intersection $\Omega_k = B_P \cap C_k^G$. If the prediction is topologically consistent, $\Omega_k$ should ideally constitute a single connected component. Conversely, if a structural fracture occurs, $\Omega_k$ will fragment into multiple sub-components.

The Fragmentation Consistency Ratio for component $k$ is:

$$\phi_k = \frac{\max_j |S_{k,j}|}{|\Omega_k| + \epsilon}, \tag{7}$$

where $\{S_{k,j}\}$ denotes the set of sub-connected components within the intersection region $\Omega_k$, and $|\cdot|$ denotes the area (number of pixels). $\phi_k$ essentially measures the dominance of the largest connected fragment. If the structure is intact, $\phi_k \approx 1$; if fractured, $\phi_k$ decreases significantly.

For each connected component $C_k^G$, a corresponding ratio $\phi_k$ is derived. Prior to aggregation, these ratios are weighted by $\omega_k$, which represents the fractional area of the component relative to the total foreground. The final $\mathcal{I}_{topo}$ score is the weighted sum of these $N$ components:

$$\mathcal{I}_{topo}(P,G) = \sum_{k=1}^N \omega_k \cdot \phi_k, \text{where } \omega_k = \frac{|C_k^G|}{\sum_{i=1}^N |C_i^G|}. \tag{8}$$

Here, $\omega_k$ explicitly couples the penalty of each independent connected component to its area. This adaptive weighting prevents minor regions from disproportionately disrupting the score, ensuring the evaluation aligns with human perception. The computational procedure for evaluating topological integrity is outlined in Algorithm 1.

### 4.3. Boundary Alignment

While $\mathcal{I}_{topo}$ captures global topology, it is relatively insensitive to local shape nuances. To complement this, we introduce $\mathcal{A}_{bdy}$ to evaluate geometric alignment via a bidirectional gradient matching strategy.

Recognizing object boundaries often exhibit a "transitional

---

**Algorithm 1** Calculation of Topological Integrity ($\mathcal{I}_{topo}$)

---

**Input:** $P, G, \tau, \epsilon$
**Output:** $\mathcal{I}_{topo}$
**Initialization:** $\mathcal{I}_{topo} \leftarrow 0; \quad B_P \leftarrow P > \tau$
Decompose $G \rightarrow \{C_k^G\}_{k=1}^N; \quad A_{total} \leftarrow \sum_{k=1}^N |C_k^G|$
**for** $k = 1$ **to** $N$ **do**
    $\Omega_k \leftarrow B_P \cap C_k^G$
    **if** $|\Omega_k| > 0$ **then**
        Compute connected components $\{S_{k,j}\}$ of $\Omega_k$
        $\phi_k \leftarrow \max_j |S_{k,j}|/(|\Omega_k| + \epsilon)$
    **else**
        $\phi_k \leftarrow 0$
    **end if**
    $\mathcal{I}_{topo} \leftarrow \mathcal{I}_{topo} + \phi_k \cdot (|C_k^G|/A_{total})$
**end for**
**Return** $\mathcal{I}_{topo}$

---

zone" of perceptually reasonable ambiguity, we design a tolerance-based soft matching approach. To avoid overfitting to pixel-perfect alignment while maintaining rigorous evaluation, we define an adaptive tolerance radius $r$:

$$r = \min(\max(1, \lfloor 0.005 \cdot \sqrt{H^2 + W^2} \rfloor), 5). \tag{9}$$

This constraint ensures the tolerance band aligns with the typical margin of human annotation error ($1 \sim 5$ pixels).

We establish symmetric tolerance bands to compute Precision and Recall independently. Let $\mathcal{M}_G^{tol}$ and $\mathcal{M}_P^{tol}$ denote the binary masks generated by dilating the edge maps of the ground truth $G$ and prediction $P$ with radius $r$, respectively. Specifically, we extract the binary edges via the morphological gradient operation (i.e., the difference between dilation and erosion) and subsequently generate the tolerance bands by dilating these edges with a disk-shaped structuring element of radius $r$. Precision ($Pre$) measures the proportion of predicted gradient energy falling within the ground truth tolerance band, while Recall ($Rec$) measures the proportion of ground truth gradient energy captured by the prediction's tolerance band:

$$Pre = \frac{\sum_{i \in \mathcal{M}_G^{tol}} \nabla P_i}{\sum \nabla P + \epsilon}, \quad Rec = \frac{\sum_{i \in \mathcal{M}_P^{tol}} \nabla G_i}{\sum \nabla G + \epsilon}, \tag{10}$$

where $\nabla P$ and $\nabla G$ represent the gradient magnitudes. Finally, $\mathcal{A}_{bdy}$ is computed as the harmonic mean:

$$\mathcal{A}_{bdy}(P,G) = \frac{2 \cdot Pre \cdot Rec}{Pre + Rec + \epsilon}. \tag{11}$$

This symmetric formulation effectively penalizes both false positive edges and false negative edges that exceed the permissible spatial tolerance.

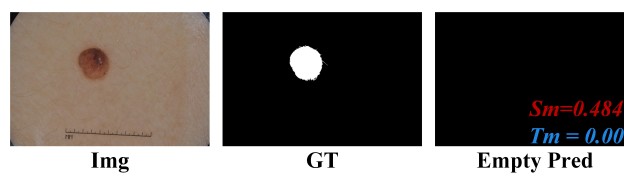

| **Img** | **GT** | **Empty Pred** |

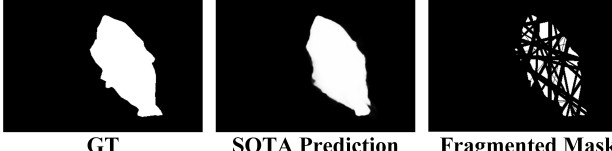

| **GT** | **SOTA Prediction** | **Fragmented Mask** |

*Figure 3.* Case of Empty Predictions. Shown are Img, GT, and all-black prediction (Empty Pred). Despite the complete failure to detect the target, $S_m$ yields a misleading score of 0.484, whereas $T_m$ correctly drops to 0.00.

*Figure 4.* Visual illustration of test samples for topological fracture sensitivity. GT: Original annotation. SOTA Prediction: Prediction map generated by ZoomNet(Pang et al., 2022). Fragmented Mask: Adversarial sample generated by applying 40 random cutting lines to the ground truth.

*Table 2.* Quantitative analysis across 200 sample pairs. We report the average scores and relative changes between SOTA predictions and Fragmented Masks. Faced with semantically meaningless fragmented masks, $T_m$ plummets by 84.3%, correctly identifying them as invalid predictions.

| Metric | SOTA | Fragmented | $\Delta\%$ |
|---|---|---|---|
| $MAE \downarrow$ | 0.045 | 0.075 | +66.7% |
| $IoU \uparrow$ | 0.843 | 0.647 | -23.3% |
| $Dice \uparrow$ | 0.901 | 0.781 | -13.3% |
| $F_m \uparrow$ | 0.908 | 0.880 | -3.1% |
| $E_m \uparrow$ | 0.938 | 0.849 | -9.5% |
| $F_\beta^w \uparrow$ | 0.882 | 0.830 | -5.9% |
| $S_m \uparrow$ | 0.898 | 0.794 | -11.6% |
| $T_m$ **(Ours)** $\uparrow$ | 0.800 | 0.126 | **-84.3%** |

Note: Unbounded boundary metrics like $95HD$ and $ASD$ are excluded as their relative changes are not directly comparable to normalized similarity metrics.

## 5. Experiments

In this section, we conduct a comprehensive validation of the proposed $T_m$. To evaluate the metric's validity, reliability, and consistency with human perception, we design four meta-evaluation experiments to benchmark $T_m$ against existing mainstream metrics across multiple tasks. Furthermore, we provide a detailed analysis of the computational efficiency of $T_m$ in Appendix B.

### 5.1. Meta-Measure 1: Robustness to Small Object Omission

In many safety-critical applications (e.g., medical tumor segmentation or camouflaged object detection), Miss Detection (completely failing to detect the target) is often considered the most severe type of error. A robust evaluation metric should impose heavy penalties on such catastrophic failures, thereby providing clear warnings to model developers. To evaluate metric sensitivity under extreme failure conditions, we selected 153 small object samples (foreground ratio $< 5\%$) from the ISIC2017(Codella et al., 2018) dataset and generated completely empty prediction maps for them.

Figure 3 presents a representative case from our collected samples. As shown, despite the prediction being a completely empty mask, $S_m$ still assigns a score of 0.484. Quantitatively, across all 153 test samples, $S_m$ achieves an average score of 0.482. This phenomenon stems from the formulation of $S_m$, which balances similarity scores between foreground and background regions. Although the model's prediction for the target is empty, it correctly classifies all background pixels, thereby inflating the score. This scoring mechanism is highly misleading and may mask severe model deficiencies in small object detection. In sharp contrast, our proposed $T_m$ yields an average score of 0.00 across all test samples. This demonstrates that $T_m$ effectively avoids such background-dominated bias, thereby offering a more rigorous and safer evaluation standard for small object detection.

### 5.2. Meta-Measure 2: Sensitivity to Topological Fracture

An excellent segmentation metric should be able to discern whether the semantic structure of the target has been

severely compromised. Many existing metrics rely heavily on pixel-level statistical features, often neglecting global topological connectivity. To verify the sensitivity of metrics to topological fracture, we designed an extreme structural fragmentation experiment.

We randomly selected 200 images from the COD10K test set. For each image, we prepared two sample sets: (1) SOTA Prediction: High-quality prediction maps generated by ZoomNet. (2) Fragmented Mask: We generated adversarial samples by applying 40 random cutting lines (width 1-10 pixels) to the Ground Truth, forcibly chopping the target into disjointed fragments (as shown in Figure 4). The topological structure of these samples is completely destroyed, stripping them of the semantic integrity required for a coherent object instance. We calculated the relative change ($\Delta\%$) of metric scores between these two groups.

The results are presented in Table 2. Surprisingly, most metrics show insensitivity to this catastrophic structural destruction. For instance, the decline in $E_m$ and $F_\beta^w$ is less than 10%, while $F_m$ drops by only 3.1%. More notably, the structure-aware metric $S_m$ still assigns a high score of 0.794, with a relative drop of only 11.6%. This implies that, from the perspective of $S_m$, these shattered fragments still constitute a high-quality prediction. This is because $S_m$ relies heavily on region-level luminance comparisons,

*Table 3.* Quantitative consistency evaluation with perceptual proxy (LPIPS). The correlation is measured using Spearman's $\rho$ and Kendall's $\tau$. **Red**, **blue**, and **green** indicate the top three results. Pixel-based metrics ($MAE$, $F_\beta^w$) naturally correlate well with LPIPS, whereas traditional structure-aware metrics ($S_m$, $95HD$) suffer significant correlation degradation. Our $T_m$ achieves top-tier perceptual consistency comparable to $MAE$ while maintaining topological awareness.

| Metric | Spearman's $\rho$ | | | | Kendall's $\tau$ | | | |
|---|---|---|---|---|---|---|---|---|
| | COD10K | DUTS | CrackSeg9k | ISIC2017 | COD10K | DUTS | CrackSeg9k | ISIC2017 |
| $IoU$ | 0.679 | 0.684 | 0.686 | 0.750 | 0.550 | 0.557 | 0.574 | 0.652 |
| $MAE$ | **0.770** | **0.791** | **0.693** | **0.784** | **0.639** | **0.667** | **0.584** | **0.686** |
| $Dice$ | 0.679 | 0.684 | 0.676 | 0.750 | 0.550 | 0.557 | 0.574 | 0.652 |
| $F_m$ | 0.571 | 0.572 | 0.568 | 0.685 | 0.452 | 0.453 | 0.465 | 0.567 |
| $E_m$ | 0.604 | 0.614 | 0.499 | 0.679 | 0.480 | 0.488 | 0.404 | 0.571 |
| $F_\beta^w$ | **0.769** | **0.792** | **0.727** | **0.776** | **0.640** | **0.669** | **0.617** | **0.675** |
| $95HD$ | 0.659 | 0.609 | 0.510 | 0.619 | 0.529 | 0.493 | 0.413 | 0.500 |
| $ASD$ | 0.709 | 0.672 | 0.625 | 0.732 | 0.580 | 0.545 | 0.525 | 0.628 |
| $S_m$ | 0.617 | 0.610 | 0.682 | 0.751 | 0.493 | 0.492 | 0.570 | 0.655 |
| $T_m$ **(Ours)** | **0.758** | **0.779** | **0.688** | **0.779** | **0.619** | **0.653** | **0.586** | **0.677** |

as long as the statistical distribution of pixels is preserved, the score remains high. In sharp contrast, our proposed $T_m$ demonstrates superior robustness. Faced with structural fragmentation, $T_m$ plummets from a SOTA score of 0.800 to 0.126, a substantial decrease of 84.3%. This drastic penalty accurately reflects the loss of semantic information, proving that $T_m$ goes beyond mere pixel counting to truly evaluate the integrity of the structure.

### 5.3. Meta-Measure 3: Consistency with Downstream Tasks

A reliable evaluation metric should serve as an accurate predictor of model utility in real-world scenarios. We identify Object Extraction as the most direct downstream application of binary segmentation. The rationale is that a higher-quality mask should yield an extracted RGB object that is perceptually closer to the ground truth.

To quantify this consistency, we adopt the Learned Perceptual Image Patch Similarity (LPIPS (Zhang et al., 2018)) as a proxy metric. Specifically, we use the predicted masks and ground-truth annotations to extract objects from the original RGB images, and feed the resulting cut-out images into LPIPS to compute perceptual similarity scores. To eliminate potential bias introduced by background colors, we further employ a dual-background strategy: similarity scores are computed separately on black and white backgrounds, and the final score is obtained by averaging the two results.

To examine the cross-domain generalization of the metrics, we conduct experiments on mainstream large-scale datasets across four domains: the DUTS(Wang et al., 2017) dataset for Salient Object Detection (test set containing 5019 images), the COD10K(Fan et al., 2020) dataset for Camouflaged Object Detection (test set containing 2026 images), the CrackSeg9k(Kulkarni et al., 2022) dataset for industrial defect detection (test set containing 1827 images), and the

ISIC2017(Codella et al., 2018) dataset for medical segmentation (test set containing 600 images).

To ensure diversity among the prediction maps, for the Salient Object Detection domain, we selected 10 SOTA models to generate predictions on the DUTS test set, including EDN(Wu et al., 2022), MENet(Wang et al., 2023), DSRNet(Song et al., 2024a), SelfReformer(Yun & Lin, 2024), ICON(Zhuge et al., 2023), ZoomNet(Pang et al., 2022), VSCode(Luo et al., 2024), MVGNet(Song et al., 2024b), VST++(Liu et al., 2024), and Samba(He et al., 2025b). Similarly, for the Camouflaged Object Detection domain, we selected 10 SOTA models to generate predictions on the COD10K test set, including RISNet(Wang et al., 2024), ZoomNet(Pang et al., 2022), VSCode(Luo et al., 2024), MVGNet(Song et al., 2024b), PRNet(Hu et al., 2024), HGINet(Yao et al., 2024), CamoFormer(Yin et al., 2024), RUN(He et al., 2025a), SENet(Hao et al., 2025), and BTDGNet(Song et al., 2025). In the domains of industrial defect detection and medical image segmentation, obtaining prediction maps is challenging due to the limited availability of open-source models. Additionally, to enhance the diversity of error types, we employed a U-Net++(Zhou et al., 2019) architecture equipped with a pre-trained VGG(Simonyan & Zisserman, 2015) backbone. The model was trained separately on the CrackSeg9k and ISIC2017 datasets. We selected model weights from 10 distinct training epochs to generate predictions on the respective test sets, thereby yielding 10 distinct types of prediction maps. This strategy aims to leverage the distinct stages of model convergence to effectively simulate a broad spectrum of failure modes, ranging from under-fitting to over-fitting, thus ensuring the robustness of the evaluation data.

We utilize Spearman's rank correlation coefficient ($\rho$)(Spearman, 1904) and Kendall's rank correlation coefficient ($\tau$)(Kendall, 1938) to measure the alignment between the metric rankings and the LPIPS proxy ranking. Spear-

man's $\rho$ primarily focuses on the overall deviation of the ranking, while Kendall's $\tau$ emphasizes the consistency of the relative order. Since the polarity of metrics varies, we report the absolute values of the correlation coefficients to provide a unified comparison of alignment strength.

Table 3 details the correlation results. We observe a clear dichotomy in existing metrics. Pixel-based metrics, particularly $MAE$ and $F_\beta^w$, exhibit strong correlations across all datasets. As expected, LPIPS determines similarity based on local texture features, inherently favoring metrics that minimize pixel-wise statistical errors. Conversely, traditional structure-aware metrics such as $S_m$, $95HD$, and $ASD$ exhibit a significant performance drop. This is attributed to their design emphasis on structural information, which consequently reduces their sensitivity to pixel-level details.

In contrast, the proposed $T_m$ bridges this gap by reconciling pixel-level perceptibility with topological consistency evaluation. While LPIPS naturally biases towards pixel statistics, $T_m$ achieves performance comparable to the purely pixel-based $MAE$ and $F_\beta^w$, consistently ranking in the top tier across all four datasets. Specifically on DUTS, the Spearman correlation coefficient ($\rho$) indicates that the gap between $T_m$ and the best-performing metric is only 0.013. More importantly, $T_m$ significantly outperforms its structural counterparts, surpassing $S_m$ and $95HD$ by 0.169 and 0.170, respectively. This provides compelling evidence that $T_m$ possesses top-tier perceptual capabilities. Crucially, while maintaining this highly competitive pixel-level perceptibility, $T_m$ incorporates topological awareness—a capability absent in other metrics.

### 5.4. Meta-Measure 4: Alignment with Human Perception

The ultimate goal of an evaluation metric is to serve as a proxy for human perception. When metric scores diverge from human judgment, their validity becomes questionable. To verify whether $T_m$ aligns better with human visual perception than existing metrics, we conducted a psychophysical experiment on samples where $T_m$ and $S_m$ exhibited significant disagreement.

We collected predictions from 20 SOTA models on the DUTS and COD10K datasets. From these, we identified Conflicting Pairs-instances where $T_m$ and $S_m$ prioritized different prediction maps as the better result for the same input. We identified 762 such pairs and randomly sampled 50 pairs (Sample IDs 1-50) as the test set for this experiment. We designed an online questionnaire using the Two-Alternative Forced Choice paradigm. The survey consisted of 50 trials. In each trial, participants were presented with four images: the Original Image, the Ground Truth, and two prediction maps (labeled A and B, corresponding to the preferences of $T_m$ and $S_m$ in randomized order). Participants were

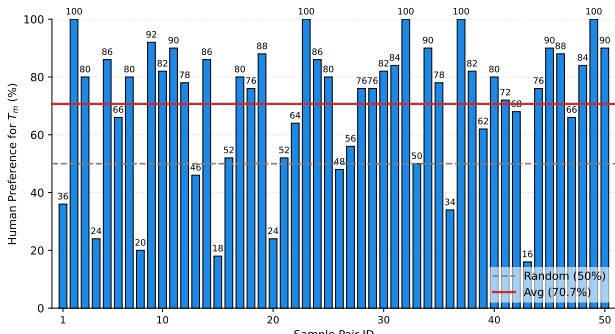

*Figure 5.* Human preference evaluation on conflicting pairs ($T_m$ vs. $S_m$). The bars indicate the percentage of participants favoring the result selected by $T_m$. The average preference rate (red solid line) is 70.7%, significantly outperforming the random chance baseline (gray dashed line, 50%).

asked to select the prediction map they perceived as most similar to the GT in terms of structure and semantics. To ensure objectivity and generality, we recruited 50 participants, comprising 20 Computer Science students and 30 individuals from unrelated fields (e.g., Finance, Medicine). The participants (32 males, 18 females), aged 20-30, had normal or corrected-to-normal vision and were naive to the experimental hypothesis.

The results are illustrated in Figure 5. The bar chart displays the proportion of human votes favoring the prediction selected by $T_m$ for each conflicting pair. Statistical analysis reveals that in cases of disagreement between $T_m$ and $S_m$, human participants favored the judgment of $T_m$ with an average probability of 70.7%. This indicates that $T_m$ demonstrates significantly higher consistency with human subjective perception, providing a more accurate reflection of true prediction quality.

## 6. Conclusion

In this paper, we critically analyzed the design principles of existing segmentation metrics, highlighted their inherent deficiency in topological perception, and proposed a novel cross-domain metric, Topology-aware measure ($T_m$). $T_m$ innovatively integrates topological consistency constraints with boundary alignment and pixel similarity to achieve a holistic and efficient evaluation. Compared to the widely used $S_m$, $T_m$ rigorously penalizes miss-detection errors and effectively identifies critical structural fragmentation. On conflicting pairs, $T_m$ achieves a high alignment rate of 70.7% with human judgment. Extensive experiments further demonstrate the capability of $T_m$ in selecting superior models for downstream tasks. In summary, we believe $T_m$ establishes a more rigorous and trustworthy benchmark for segmentation tasks, encouraging the community to prioritize topological correctness in future research.

# Acknowledgements

This work was supported by the National Natural Science Foundation of China under Grant 62273268.

# Impact Statement

This paper presents work whose goal is to advance the field of Machine Learning. There are many potential societal consequences of our work, none which we feel must be specifically highlighted here.

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

# A. Sensitivity Analysis of Hyper-parameters

As defined in Eq. (5) of the main paper, the proposed metric $T_m$ is formulated as:

$$T_m(P, G) = \mathcal{S}_{pix} \times [\alpha \mathcal{I}_{topo} + (1 - \alpha)\mathcal{A}_{bdy}], \tag{12}$$

where the hyper-parameter $\alpha$ governs the trade-off between the penalty for topological violations ($\mathcal{I}_{topo}$) and the penalty for boundary misalignment ($\mathcal{A}_{bdy}$). To ensure that $\alpha$ is calibrated to reflect human visual judgment, we conducted a Psychophysical Perception Experiment.

## A.1. Experimental Setup

We designed a subjective user study involving 50 participants. The visual stimuli consisted of a source RGB image and two distinct failure cases:

- $P_{bdy}$ (Boundary Noise): A prediction with jagged edges but correct topology.

- $P_{topo}$ (Topological Fracture): A prediction with smooth edges but a fractured structure.

To further control variables, $P_{bdy}$ and $P_{topo}$ were generated to have nearly identical pixel-level errors, with the difference in Mean Absolute Error (MAE) kept below 0.002. To establish a unified evaluation scale, we provided the Ground Truth (GT) as the standard for a perfect score (10 points) and an Empty Prediction as the anchor for the lowest score (0 points). Participants were asked to rate $P_{bdy}$ and $P_{topo}$ based on structural integrity using integers from 0 to 10. The interface and question setup are illustrated in Figure 6.

It is important to note that we deliberately employed a single representative case for this evaluation. The rationale is that constructing a large-scale dataset inevitably introduces uncontrolled variance due to data complexity and inconsistent error severities across samples, which would cause the averaged scores to fail in accurately reflecting human sensitivity to boundary noise versus topological fracture. By using a carefully designed single exemplar, we effectively eliminate confounding variables such as object semantics, scale, and contrast. This design achieves strict variable control, ensuring that participants' ratings purely reflect the perceptual trade-off between topological integrity and boundary precision.

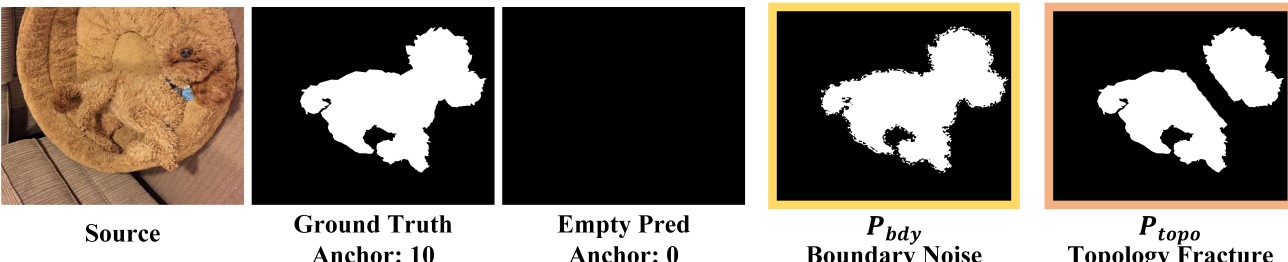

**Q: Compared to the perfect Ground Truth (10), how well does each result preserve the structure of the object?**

| Source | Ground Truth Anchor: 10 | Empty Pred Anchor: 0 | $P_{bdy}$ Boundary Noise | $P_{topo}$ Topology Fracture |

*Figure 6.* Visual interface used in the psychophysical experiment. Participants were presented with the Source Image, Ground Truth (Anchor Score = 10), and Empty Prediction (Anchor Score = 0). They were asked to rate two specific failure modes: $P_{bdy}$ (Boundary Noise) and $P_{topo}$ (Topology Fracture), which have nearly identical pixel-level errors.

We analyzed the survey data using the *Perceptual Preference Ratio* ($R$), defined as the score of $P_{bdy}$ divided by the score of $P_{topo}$ ($R = Score_{bdy}/Score_{topo}$). Specifically, we calculated the ratio for each participant individually. The statistical results yield a mean preference ratio of 1.295. Furthermore, we determined the standard deviation ($\sigma$) of these individual ratios to be 0.136. We define the interval $[1.295 - \sigma, 1.295 + \sigma]$ as the Human Perception Range, representing the reasonable fluctuation of human sensitivity to structural errors.

We then simulated the response of $T_m$ to these two cases by sweeping $\alpha$ from 0.1 to 0.9. The raw scores are detailed in Table 4. To visualize the alignment, we plotted the ratio of $T_m$ scores ($T_m(P_{bdy})/T_m(P_{topo})$) against the human baseline, as shown in Figure 7.

*Table 4.* $T_m$ scores of the test cases evaluated under varying $\alpha$. We evaluated the two controlled failure cases ($P_{bdy}$ and $P_{topo}$) using $T_m$ with $\alpha$ ranging from 0.1 to 0.9. Observe that as $\alpha$ increases, the metric penalizes the topological fracture ($P_{topo}$) more severely.

| Hyper-param $\alpha$ | 0.1 | 0.2 | 0.3 | 0.4 | 0.5 | 0.6 | 0.7 | 0.8 | 0.9 |
|---|---|---|---|---|---|---|---|---|---|
| $P_{bdy}$ | 0.662 | 0.691 | 0.720 | 0.749 | 0.778 | 0.807 | 0.836 | 0.865 | 0.894 |
| $P_{topo}$ | 0.814 | 0.789 | 0.765 | 0.741 | 0.716 | 0.691 | 0.667 | 0.643 | 0.618 |

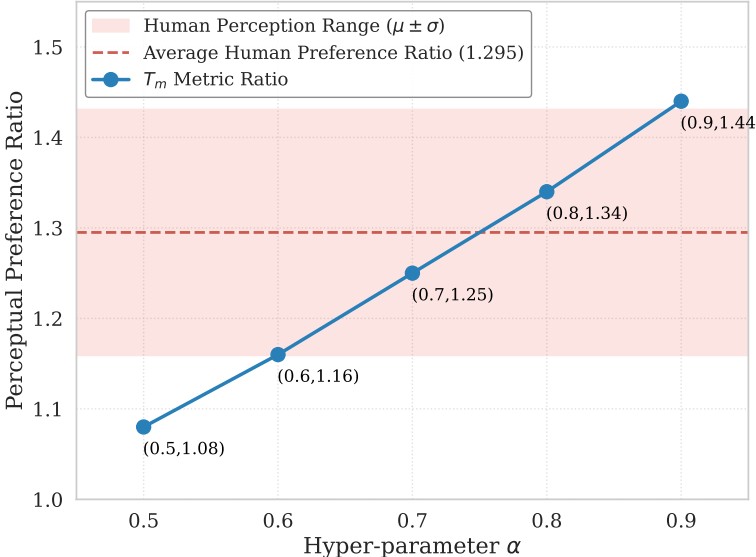

*Figure 7.* Parameter sensitivity analysis aligned with human perception. The blue line tracks the ratio of $T_m$ scores ($Score_{bdy}/Score_{topo}$) across different $\alpha$ values. The red dashed line represents the average human preference ratio (1.295), with the red transparent band highlighting the Human Perception Range (defined as mean $\pm\sigma$, where $\sigma = 0.136$, $N = 50$).

### A.2. Results and Analysis

As illustrated in Figure 7, when $\alpha$ is set to 0.7 or 0.8, the metric's ratios fall well within the Human Perception Range. Remarkably, both values exhibit nearly identical deviations from the mean human preference ratio. However, an evaluation metric serves as a guide for model development and should ideally enforce a higher standard. We aim to impose a stricter penalty on topological errors to prevent models from overfitting to boundary details at the expense of structural correctness. Therefore, we adopt $\alpha = 0.8$ ($R = 1.34$) as the default setting. This value not only aligns with the stricter end of the human perception spectrum but also provides a robust safety margin for topological integrity.

Beyond alignment with human perception, we strongly advocate for setting $\alpha > 0.5$ from the perspective of model optimization. In segmentation tasks, boundary errors and topological errors present distinct challenges in terms of optimization difficulty. Boundary errors typically involve pixel-level fine-tuning, where correcting these misalignments is a relatively straightforward process achievable through standard gradient descent strategies. Conversely, topological errors represent a fundamental semantic failure. Such errors often indicate that the model is trapped in a stubborn local minimum. Resolving these structural defects requires the model to capture global context, a task significantly more demanding than mere edge refinement. Therefore, the evaluation metric must enforce a stronger penalty signal for structural defects. This prevents the model from settling for easy boundary improvements while neglecting hard topological failures.

## B. Computational Efficiency Analysis

### B.1. Experimental Setup

To evaluate the practical applicability of the proposed metric, we conducted a comprehensive computational efficiency benchmark. The experiments were performed on a standard Intel(R) Core(TM) i7-13700F CPU @ 2.90GHz using a single thread. All metrics were implemented in Python 3.8, utilizing the NumPy library for vectorized operations. Specifically, for distance-based metrics ($95HD$, $ASD$), we employed the optimized distance transform routines from the SciPy library. To

*Table 5.* Computational efficiency benchmark results. Our $T_m$ achieves 24.6 FPS. Compared to other complex structural/distance metrics (e.g., $F_\beta^w$, $95HD$, $ASD$), our metric demonstrates a significant advantage in computational efficiency while simultaneously providing topological awareness.

| Metric | Time (ms/img) ↓ | FPS ↑ |
|---|---|---|
| $IoU$ | 6.61 | 151.3 |
| $Dice$ | 6.80 | 147.0 |
| $MAE$ | 9.34 | 107.1 |
| $F_m$ | 13.51 | 74.0 |
| $E_m$ | 15.22 | 65.7 |
| $S_m$ | 23.91 | 41.8 |
| $95HD$ | 74.22 | 13.5 |
| $ASD$ | 74.55 | 13.4 |
| $F_\beta^w$ | 91.05 | 11.0 |
| $T_m$ (Ours) | 40.69 | 24.6 |

ensure fairness and reproducibility, the implementations of all baseline metrics strictly adhere to their original definitions and align with widely adopted community open-source benchmarks. We measured the average inference time per image (ms/img) and Frames Per Second (FPS) on the COD10K test set (Resolution: $352 \times 352$). To facilitate reproducibility, the complete benchmarking script, including the implementations of all comparison metrics and environment configurations, is provided in our supplementary material.

### B.2. Results and Analysis

The benchmark results are presented in Table 5. Despite incorporating topological awareness, $T_m$ maintains a highly competitive calculation speed of 40.69 ms/img (24.6 FPS). Notably, $T_m$ significantly outperforms other advanced structural and distance-based metrics. It is approximately $2.2\times$ faster than $F_\beta^w$ and $1.8\times$ faster than distance-transform-based metrics such as $95HD$ and $ASD$. With a processing speed of nearly 25 FPS on a standard CPU, $T_m$ satisfies the real-time evaluation requirements for most offline validation scenarios.

## C. Decoupling Characteristics and Error Diagnosis Potential

Existing evaluation metrics (e.g., IoU) typically provide a holistic scalar score. While this facilitates coarse model ranking, it often masks the root causes of performance degradation: does the score drop stem from fundamental semantic failure, or merely from insufficient boundary refinement ?

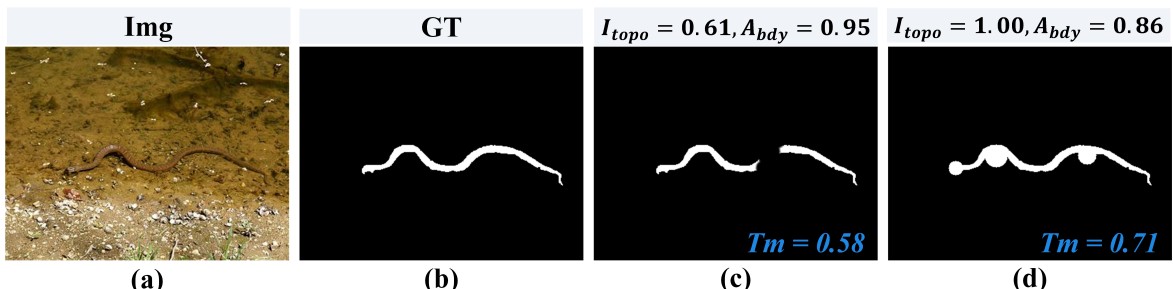

*Figure 8.* Error diagnosis based on the $T_m$ metric. (a) Source RGB image. (b) Ground Truth (GT). (c) Structural Fracture Case. (d) Boundary Noise Case. By observing the sub-term scores, researchers can rapidly distinguish the nature of the errors.

As discussed in Appendix A, these two types of errors exhibit significant differences from the perspective of model optimization: boundary errors usually only involve pixel-level fine-tuning; whereas topological errors often indicate that the model is trapped in a stubborn local minimum, representing a failure of global semantic capture, which is extremely difficult to rectify. Therefore, an ideal metric should not only reflect performance levels but also possess the capability of error attribution to guide researchers in targeted optimization.

Benefiting from the modular design of $T_m$, our metric inherently possesses decoupling characteristics. By independently observing the values of $\mathcal{I}_{topo}$ and $\mathcal{A}_{bdy}$, users can clearly diagnose the "bottlenecks" of the model. To validate this characteristic, we present a typical diagnosis case in Figure 8. We selected a source RGB image, the Ground Truth (GT), and two representative prediction results for analysis: As shown in Figure 8 (c), the prediction fails to preserve target connectivity, resulting in distinct fractures. Under traditional metrics, we only observe a relatively low score ($T_m = 0.58$). However, by examining the sub-terms of $T_m$, we discover that its boundary score is extremely high ($\mathcal{A}_{bdy} = 0.95$), while the topological score is significantly low ($\mathcal{I}_{topo} = 0.61$). This data signature precisely pinpoints the root cause of the error: the model excels at local boundary processing but suffers from severe defects in global structural modeling. This suggests that developers should prioritize optimizing the model's contextual capture capability (e.g., enlarging the receptive field or introducing attention mechanisms) rather than adjusting the boundary loss.

In summary, $T_m$ serves not only as an evaluation tool but also as a diagnostic tool. This fine-grained error attribution capability enables researchers to distinguish topological failures from boundary noise, thereby allowing for the formulation of more efficient model improvement strategies.

