# OpenReview forum: "T-measure: A Topology-Consistent Metric for Binary Segmentation"
_ICML.cc/2026/Conference — ICML 2026 regular_

### Official Review · Reviewer_zLYP · 2026-03-10

**Soundness:** 2
**Presentation:** 2
**Significance:** 2
**Originality:** 2
**Overall Recommendation:** 4
**Confidence:** 3

**Summary:**

This paper proposes a novel topological consistency metric called T-measure (Tm), specifically designed for binary segmentation tasks. To address the shortcomings of existing metrics in capturing structural integrity and handling small objects, this metric integrates a pixel-level fuzzy Jaccard index, a topological integrity penalty term, and a boundary alignment assessment term.

**Compliance With Llm Reviewing Policy:**

Affirmed.

**Final Justification:**

Thank you very much for your detailed response to my query. This has resolved my concerns and I have decided to adjust my score to 4.

**Key Questions For Authors:**

1. To what extent is the topological integrity penalty term affected by the fixed binarization threshold (0.5)? If the model's probability output is generally low or poorly calibrated, how can the stability of this metric be guaranteed?
2. Please clearly explain what the strict mathematical or experimental basis is for each constant in the boundary tolerance radius formula.
3. When dealing with real targets that are naturally composed of multiple small disconnected fragments, will overemphasizing the dominance of the largest connected component lead to excessive punishment?
4. Given that the human preference experiment only selected 50 pairs from 762 conflicting samples for testing, can the authors provide a more detailed statistical significance analysis to prove that the current sample size is sufficient to support the global conclusion of "highly consistent with human intuition"?

**Limitations:**

The paper's discussion on its own limitations remains weak, particularly in the aspects of complex application scenarios and the robustness of hyperparameters. The additional introduction of connected domain calculation and morphological gradient operation significantly increases the computational cost of this metric, which essentially limits its application potential in large-scale online benchmarking systems that require real-time feedback.

**Strengths And Weaknesses:**

Strengths:

1. It accurately pointed out the shortcomings of existing metrics (such as S-measure) in handling topological breaks and small target detection.
2. The proposed metric has a modular decoupling feature, enabling independent assessment of boundary alignment and topological integrity respectively.

Weaknesses:

1. Although claiming to effectively evaluate the original probability map, the topological completeness term relies on a fixed binarization threshold (set at 0.5).
2. This hard-threshold-based connected component calculation method may lead to extremely unstable evaluation results when the predicted probability distribution is uneven.
3. Small sample sizes weaken the statistical persuasiveness of the core claim that this metric is highly consistent with human subjective perception.
4. The hard-coded design lacks a sufficient theoretical basis and may lack generalization ability on high-resolution or low-resolution images of different resolutions.

---

> ### Author Rebuttal · Authors · 2026-03-30
>
> We thank the reviewer for the constructive comments.
>
> **1:Binarization Threshold(W1,W2,Q1)**
>
> (1)Constrained by the objective requirements of topological feature extraction, $I_{topo}$ relies on binarization operations, but both $S_{pix}$ and $A_{bdy}$ in the metric can directly process continuous probability distributions, retaining the soft-evaluation capability of $T_m$ to a certain extent.
>
> (2)Regarding the impact of the binarization threshold on evaluation stability, we selected the prediction results of 10 SOTA models on the COD10K (consistent with the settings in the Meta Measure 3), calculated the scores of each model under different thresholds, and ranked them. We computed the Spearman rank correlation coefficients between the sequences.
>
> Tab.1: Spearman correlation matrix
> |Threshold|0.3|0.4|0.5|0.6|0.7|
> |---|---|---|---|---|---|
> |0.3|1.00|1.00|1.00|1.00|0.99|
> |0.4|1.00|1.00|1.00|1.00|0.99|
> |0.5|1.00|1.00|1.00|1.00|0.99|
> |0.6|1.00|1.00|1.00|1.00|0.99|
> |0.7|0.99|0.99|0.99|0.99|1.00|
>
> All correlation coefficients under all thresholds are greater than 0.99, showing that $T_m$ has strong robustness to the setting of the binarization threshold.
>
> (3)In extreme cases where model probability outputs are generally low, we provide an adaptive thresholding option that automatically adjusts the binarization baseline based on the input probability distribution, avoiding evaluation failures(see line 35 of t_measure.py in our code).
>
> **2:Setting of the Boundary Tolerance Radius(W4,Q2)**
>
> The setting of the range constant conforms to general computer vision evaluation standards. The official summary paper of the PASCAL VOC semantic segmentation challenge (Ref.[1]) points out that a range of 3 to 5 pixels around the true boundary is set as an ignored boundary region to eliminate annotation uncertainty. Based on this, the tolerance band of the $T_m$ metric is restricted to 1 to 5 pixels.
>
> Based on a study of annotators, Ref. [2] proposes using a fixed percentage of the image diagonal length (0.02 in its benchmark setting) to dynamically calculate the boundary tolerance distance, and $T_m$ adopts this idea and sets a stricter limit of 0.005, ensuring scale invariance across different resolutions.
>
> [1]PASCAL VOC Challenge A Retrospective,IJCV,2015.
>
> [2]Boundary IoU,CVPR,2021.
>
> **3:Human Perceptual Experiments(W3,Q4)**
>
> (1)Due to the high similarity of prediction maps, large-scale ranking experiments involving multiple prediction maps significantly increase the burden on subjects, leading to unreliable results. Mainstream evaluation metrics $S_m$ and $E_m$ both fully considered the above limitations. $S_m$ selected 45 subjects and 50 pairs of samples as test materials. $E_m$ only invited 10 subjects to rank the prediction maps. $T_m$ followed the design of the benchmark $S_m$, selecting 50 pairs of conflicting samples and inviting 50 subjects for testing.
>
> (2)We added two statistical tests.
>
> * Binomial distribution test based on the total number of independent votes. The total votes $n=2500$, random probability $p_0=0.5$. The $Z$ statistic is 20.72, corresponding to $p<0.001$.
>
> * For the one-sample $t$-test based on sample pairs, the sample size is set to $n=50$. The mean support rate is 70.72% with a 3.45% standard error, $t$ statistic is 6.00, $p<0.001$ with 49 degrees of freedom.
>
> These tests prove statistical significance, indicating the current sample size supports the conclusion of consistency with human intuition.
>
> **4:Analysis of Fragmented Targets(Q3)**
>
> When processing disconnected targets, $T_m$ will not cause over penalization. As stated in Sec. 4.2 of the original manuscript, the metric first decomposes the true foreground into $N$ independent connected components, independently calculates the fragment consistency ratio $\phi_k$ within each connected component, and finally performs weighted aggregation based on area proportions. $T_m$ consistently focuses on the structural integrity within independent targets, making it compatible with highly discrete scenarios such as cell segmentation and dense crowd segmentation.
>
> **5:Limitations**
>
> (1)$T_m$ is not applicable to multi-class tasks. Designing a joint penalty mechanism for inter-class topological conflicts is one of our future research directions.
>
> (2)We tested the hyperparameter $\alpha$. The results indicate that adjusting $\alpha$ within the interval $[0.5, 0.9]$ across different domains maintains evaluation stability, with ranking correlations consistently greater than 0.9. This demonstrates the strong cross-domain generalization and robustness of $\alpha$. (Detailed experimental results are provided in our response to Reviewer U8wA.)
>
> (3)Computational efficiency has been provided in Appendix B of the original manuscript. Despite the introduction of connected component analysis, the computational efficiency of $T_m$ (24.6 FPS) remains significantly superior to the widely used 95HD (13.5 FPS), ASD (13.4 FPS), and $F_{\beta}^{w}$ (11.0 FPS) metrics.

---

> > ### Author Rebuttal · Reviewer_zLYP · 2026-04-03
> >
> > Thank you very much for your detailed response to my question. This has resolved my concerns and I have decided to adjust my score to 4.

---

> > > ### Author Response · Authors · 2026-04-04
> > >
> > > Thank you for your positive feedback on our response. We also sincerely appreciate the time and effort you devoted to reviewing our manuscript.

---

### Official Review · Reviewer_fyWF · 2026-03-11

**Soundness:** 3
**Presentation:** 3
**Significance:** 2
**Originality:** 3
**Overall Recommendation:** 4
**Confidence:** 3

**Summary:**

This paper proposes a novel evaluation metric for binary segmentation called T-measure, aiming to address the limitations of existing metrics (such as IoU, Dice, and S-measure) in measuring topological consistency. The authors appear to explore the concept of integrating the Fuzzy Jaccard Index, a topological integrity term based on connected component analysis, and a boundary alignment term. T-measure is designed to effectively penalize severe structural fractures and missed detections. Experimental results demonstrate that in extreme failure scenarios, T-measure shows a more significant penalty than traditional metrics and exhibits a higher alignment with human assessment.

**Compliance With Llm Reviewing Policy:**

Affirmed.

**Final Justification:**

After reading the rebuttal, all my concerns have been resolved.

**Key Questions For Authors:**

- The calculation of T-measure appears significantly more complex than current mainstream metrics. Could you provide a comparison of its computation time relative to other metrics?
- Which specific domains, object types, or segmentation objectives are most suitable for T-measure, and which are not?

**Limitations:**

The paper does not sufficiently explore its limitations. It is suggested that the authors provide an analysis from the perspective of which fields or tasks T-measure is particularly well-suited for versus those where it may be less effective.

**Strengths And Weaknesses:**

Strengths:
- The paper introduces T-measure, a new metric for binary segmentation. Compared to S-measure, experiments demonstrate its advantages in alignment with human perception.
- T-measure effectively focuses on the intrinsic challenges of the segmentation field (topological consistency), which is quite inspiring.

Weaknesses:
- Regarding the evaluation of alignment between T-measure and human perception, the comparison is only conducted against S-measure, lacking comparisons with other relevant metrics. Furthermore, there is a lack of experiments testing the alignment between human ranking of multiple predictions and the ranking produced by T-measure. Consequently, the evidence for its superiority feels insufficient.
- This metric may not be suitable for all segmentation scenarios. It is recommended that the authors provide a more in-depth discussion on its applicable scope.

---

> ### Author Rebuttal · Authors · 2026-03-29
>
> We thank the reviewer for the constructive comments.
>
> **1:Human Perceptual Consistency(W1)**
>
> (1) In the human perceptual consistency experiment, $T_m$ is compared with $S_m$ because $S_m$ is the consensus benchmark in the field of structure-aware evaluation. Our core motivation is to verify whether $T_m$ can exhibit human visual consistency surpassing $S_m$ when processing complex structures. Comparisons with other metrics are primarily demonstrated through LPIPS ranking experiments. LPIPS itself is trained on large-scale human perception data and is a widely recognized deep learning metric for low-level visual perception in academia. The large-scale experiment in Meta Measure 3 of the original manuscript reflects the performance of 10 metrics in human perceptual consistency to a certain extent, demonstrating the consistency of $T_m$ with human visual perception benchmarks.
>
> (2) Experimental Design for Human Perceptual Consistency. Requiring subjects to rank multiple highly similar prediction maps from state-of-the-art models significantly increases cognitive burden. Moreover, evaluating too many samples causes subjects to lose focus, ultimately leading to unreliable and unstable choices. Mainstream evaluation metrics $S_m$ and $E_m$ both fully considered the above limitations. $S_m$ (Ref.[1]) selected 45 subjects and 50 pairs of samples as test materials. $E_m$ (Ref.[2]) only invited 10 subjects to rank the prediction maps. $T_m$ followed the design of the benchmark literature $S_m$, selecting 50 pairs of conflicting samples and inviting 50 subjects for testing. A final support rate of 70.7% was obtained, which has a significant advantage over the 63.69% support rate of $S_m$.
>
> [1] Structure Measure A New Way to Evaluate Foreground Maps, ICCV, 2017.
>
> [2] Enhanced Alignment Measure for Binary Foreground Map Evaluation, IJCAI, 2018.
>
> **2:Computational Efficiency(Q1)**
>
> The computational efficiency comparison between $T_m$ and other metrics have been reported in Appendix B of the original manuscript. The results show that despite the introduction of connected component analysis and morphological operations, the computational efficiency of $T_m$ (24.6 FPS) is still significantly higher than the widely used 95HD(13.5 FPS), ASD (13.4 FPS), and $F_{\beta}^{w}$ (11.0 FPS) metrics.
>
> **3:Applicability and Limitations(W2, Q2, Limitations)**
>
> (1) General evaluation capability. $T_m$ is not limited to specific domains. The pixel similarity term $S_{pix}$ can measure the pixel classification accuracy of prediction maps, ensuring its generality for cross-domain binary segmentation evaluation. Both $S_{pix}$ and $A_{bdy}$ in the metric can directly process continuous probability distributions, retaining the ability to evaluate probability maps to a certain extent.
>
> (2) Evaluation advantages and diagnostic capabilities in specific scenarios. $T_m$ introduces two independent sub-items $I_{topo}$ and $A_{bdy}$, endowing the metric with the ability to perceive high-level target structures. Therefore, $T_m$ is more suitable for tasks sensitive to structural information, such as blood vessel segmentation, electronic component crack detection, and camouflaged animals with blurred boundaries.
>
> (3) Discussion on limitations.
>
> $T_m$ cannot be used as a loss function because its calculation process is non-differentiable. The positioning of $T_m$ is to provide objective quality ranking and deep error diagnosis for models. If expanded into a differentiable form, continuous algebraic approximations must be used to replace strict discrete topological calculations, which would significantly increase computational complexity and introduce approximation errors. Breaking through these limitations is one of the future research directions.
>
> $T_m$ is currently not applicable to multi-class segmentation tasks. Although the $T_m$ metric can be naturally extended to a multi-class segmentation evaluation system through an inter-class macroscopic averaging strategy, we believe that true multi-class topology perception is not simply the summation of single-class topologies. In multi-class scenarios, complex inter-class topological relationships often exist. Currently, $T_m$ lacks a joint penalty mechanism designed for these inter-class topological conflicts, which is an important direction for future work.
>
> $T_m$ cannot be applied to large-scale online benchmark systems requiring real-time feedback. The core design goal of $T_m$ is to provide advanced topological perception capabilities surpassing conventional pixel-level metrics. Therefore, connected component analysis and morphological operations are introduced, inevitably increasing computational costs. Appendix B of the original manuscript provides detailed computational efficiency test results, proving that $T_m$ can satisfy quality assessment tasks, but a gap in real-time performance remains compared to pixel-level metrics.

---

> > ### Author Rebuttal · Reviewer_fyWF · 2026-04-03
> >
> > Thanks to the authors for carefully answering my questions. After reading the rebuttal, my concerns have been fully resolved. Hence, I will improve my recommendation.

---

> > > ### Author Response · Authors · 2026-04-04
> > >
> > > Thank you for your positive feedback on our response. We also sincerely appreciate the time and effort you devoted to reviewing our manuscript.

---

### Official Review · Reviewer_U8wA · 2026-03-13

**Soundness:** 3
**Presentation:** 3
**Significance:** 2
**Originality:** 3
**Overall Recommendation:** 4
**Confidence:** 3

**Summary:**

The authors propose a novel evaluation metric for binary image segmentation termed the Topology-aware measure, designed to quantify structural integrity and geometric precision. The metric functions as a unified framework that integrates three synergistic components: a pixel-level similarity term based on fuzzy set theory, a topological integrity term that employs connected component analysis to penalize structural fragmentation, and a boundary alignment term that uses an adaptive tolerance-based gradient matching strategy. To validate the metric, the authors establish a Meta-Measure evaluation framework and benchmark the proposed method against nine mainstream metrics across four distinct segmentation domains

**Compliance With Llm Reviewing Policy:**

Affirmed.

**Final Justification:**

Most of my concerns are resolved. I keep my score as weak accept.

**Key Questions For Authors:**

Please carefully answer these questions, the reviewer may change the score according to the rebuttal.

1. Since there are also some existing metrics evaluating the topological integrity and boundary alignment, what makes the proposed metric better than those works in evaluating these aspects respectively ?  To the reviewer, this manuscript looks like combining different metrics and propose a combined overall score. However, we can just evaluate them separately to relect the ability flaw in different dimention. Why should we use such a combined score instead of using exisiting metrics to form a fine-grained quality evaluation?

2. Does the proposed metric reveal new flaw in existing SOTA segmentation methods (compared with existing methods, especially topology-aware methods like [1] and [2]) ?

3. Can the proposed metric maitain its effiency on multi-class segmentation tasks and high-resolution images, like Cityscapes (2048x1024)?

[1] Berger, Alexander H., et al. "Pitfalls of topology-aware image segmentation." International Conference on Information Processing in Medical Imaging. Cham: Springer Nature Switzerland, 2025.

[2] He, Hongliang, et al. "Toposeg: Topology-aware nuclear instance segmentation." Proceedings of the IEEE/CVF International Conference on Computer Vision. 2023.

**Limitations:**

no. please refer to the weakness and key questions

**Strengths And Weaknesses:**

## Strength
1. The writing is clear and easy to follow.
2. The introduced $I_{topo}$ is reasonable for topology-aware evaluation.

## Weaknesses

1. The proposed metric $T_m$ integrates disparate evaluation dimensions, namely the pixel-wise overlap ($S_{pix}$), topological integrity ($I_{topo}$), and boundary alignment ($A_{bdy}$), into a single scalar value. This aggregation leads to "dimensional collapse", where the nuanced performance characteristics of different segmentation models are obscured. The reviewer believes the quality of prediction should be evaluated in different dimentions to show different ability independently. A combined metric is not better than a set of fine-grained metrics.

2. The formulation of $T_m$ relies on a defined weight ($\alpha=0.8$) to balance topological integrity and boundary precision. This hyper-parameter is derived from a psychophysical experiment based on a single, controlled failure case, which lacks sufficient theoretical justification for generalization across diverse domains (e.g., medical imaging vs. industrial defect detection). There is no comprehensive robustness analysis provided for $\alpha$ across varying segmentation methods.

3. While the paper introduces a new metric, it fails to dive in to analyze existing state-of-the-art (SOTA) segmentation models to reveal previously undetected defects. The authors treat the metric primarily as a ranking tool rather than a diagnostic lens. The study does not convincingly demonstrate how $T_m$ changes our understanding of current segmentation algorithms. Without a detailed breakdown or qualitative analysis of how SOTA segmentation models fail under the $T_m$ compared to traditional metrics (e.g., IoU, Dice, $S_m$, boundary IoU, boundary F-score), the metric serves more as a "post-hoc ranking convenience" than an insightful contribution to the segmentation model evaluation.

4. The proposed framework is constrained to binary segmentation, with no discussion on its extension to multi-class semantic segmentation. In multi-class scenarios, the metric would encounter severe challenges, including inter-class topological interference, complex class-boundary ambiguity, and non-linear increases in computational overhead due to multiple connected component analyses.

5. The computation of $T_m$ involves morphological gradient operations and connected component analysis (CCA), which are inherently non-differentiable and computationally intensive, especially for high-resolution imagery. Unlike metrics that are easily integrated into training pipelines, $T_m$ remains a static post-processing evaluation tool. The lack of a differentiable approximation or an optimized GPU-based implementation limits its utility in the modern deep learning workflow, where real-time validation and loss-based optimization are critical.

---

> ### Author Rebuttal · Authors · 2026-03-29
>
> We thank the reviewer for the constructive comments.
>
> **1:Unified Evaluation and Advantages of the Sub-items(W1,Q1)**
>
> (1) $T_m$ provides a unified framework covering multiple dimensions to evaluate the high-level perceptual capabilities of models. The sub-items are not simply concatenated but form a constrained synergistic relationship through a multiplicative weighting mechanism, which avoids the problem of errors being diluted. Additionally, compared with the existing structure-aware metric $S_m$, $I_{topo}$ more accurately identifies topological fracture phenomena and fixes evaluation failures in small object scenarios. Compared with the existing boundary-aware metric $95HD$, $A_{bdy}$ effectively alleviates the numerical amplification effect caused by isolated noise points.
>
> (2) In large-scale benchmark tests, multiple independent metrics often produce ranking conflicts, leading to an inability to form stable conclusions. $T_m$ provides a unified ranking standard, and $T_m$ adopts an explicitly decomposable design where all three sub-items can be independently reported for error diagnosis, preserving fine-grained evaluation capabilities.
>
> **2:Revealing Defects in Topology Models(W3,Q2)**
>
> (1) Reference [1] points out that existing topology-aware models may suffer from topological overfitting phenomena when optimizing connectivity loss by generating extremely thin connections in fractured regions to maintain connectivity. Pure pixel metrics cannot perceive such anomalies, and pure topological metrics give perfect evaluation scores because the connection count meets the standard. Through multi-dimensional synergistic constraints, $T_m$ imposes a reasonable penalty on such errors, revealing the potential defects of topology-aware models.
>
> (2) We generated 200 pairs of simulated samples based on the CrackSeg9K dataset, including fractured samples PredA and topological overfitting samples PredB. The average scores are shown in Tab.1.
>
> Tab.1: Simulation results
> |Sample|PredA|PredB|
> |:---|:---|:---|
> |Betti Error $\downarrow$|1.00|0.00|
> |$T_m$ $\uparrow$|0.54|0.69|
>
> The pure topological metric fails to identify the topological overfitting phenomenon, erroneously giving an evaluation of zero error. $T_m$ not only identifies topological fractures but also imposes a reasonable penalty on topological overfitting. More results can be found in Fig.1 under the TopologicalOverfitting directory of our code repository.
>
> [1] Pitfalls of topology aware image segmentation, IPMI, 2025.
>
> **3:Hyperparameter Generalization and Robustness(W2)**
>
> To verify hyperparameter robustness, we calculate the ranking correlation of 10 models within each of the four domains under different $\alpha$ values. Settings match Meta Measure 3. Using $\alpha=0.8$ as the baseline, we compute Spearman rank correlations. The results are shown in Tab.2.
>
> Tab.2: Hyperparameter generalization analysis
> |Dataset|0.5|0.6|0.7|0.9|
> |:---|:---|:---|:---|:---|
> |DUTS|1.00|1.00|1.00|1.00|
> |COD10K|1.00|1.00|1.00|1.00|
> |CrackSeg9K|0.96|0.98|1.00|0.99|
> |ISIC2017|0.93|0.96|0.99|0.99|
>
> All correlations across all domains exceed 0.9, presenting an extremely strong positive correlation. This proves researchers can obtain reliable performance rankings in different domains without parameter adjustment.
>
> **4:Limitations of the Metric(W4,W5)**
>
> (1) $T_m$ can be extended to a multi-class segmentation evaluation system through an inter-class macroscopic averaging strategy, and $T_m$ can be used to independently evaluate each class. However, we believe that true multi-class topology perception is not simply the summation of single-class scores. In multi-class scenarios, complex inter-class topological relationships often exist. Currently, $T_m$ has not designed a joint penalty mechanism for these inter-class topological conflicts, which is one of our future directions.
>
> (2) The calculation process of $T_m$ is non-differentiable. If the metric is forcibly designed into a differentiable form, continuous algebraic approximations must be adopted, which would significantly increase computational complexity and introduce approximation errors. The positioning of $T_m$ is to provide objective quality ranking and deep error diagnosis for models, so we did not choose to sacrifice computational efficiency for the sake of differentiability.
>
> **5:Computational Efficiency on High-Resolution(Q3)**
>
> As described, $T_m$ is extended to multi-class tasks by evaluating each category independently and calculating the average (the same applies to other metrics). Per-class FPS and per-image FPS are reported on the Cityscapes dataset($2048 \times 1024$).
>
> Tab.3: Efficiency comparison
> |Metric|FPS(Class)|FPS(Image)|
> |:---|:---|:---|
> |$S_m$|11.2|0.67|
> |$95HD$|3.4|0.31|
> |$ASD$|3.4|0.31|
> |$F_{\beta}^{w}$|3.1|0.28|
> |$T_m$|8.0|0.41|
>
> The computational efficiency of $T_m$ on high-resolution images remains superior to the 95HD, ASD, and $F_{\beta}^{w}$ metrics.

---

> > ### Author Rebuttal · Reviewer_U8wA · 2026-04-03
> >
> > Most of my concerns are resolved. I tend to keep my score.

---

> > > ### Author Response · Authors · 2026-04-04
> > >
> > > Thank you for your positive feedback on our response. We also sincerely appreciate the time and effort you devoted to reviewing our manuscript.

---

### Decision · Program_Chairs · 2026-04-30

**Decision:**

Accept (regular)

**Comment:**

This paper introduces a topology-aware evaluation metric (T-measure) for binary image segmentation, aiming to jointly capture pixel-wise accuracy, structural integrity, and boundary alignment. The topic is relevant and timely, particularly for applications where structural consistency is critical.

There is general agreement among the reviewers that the proposed metric is technically sound and reasonably well-motivated. The integration of multiple evaluation dimensions is seen as a practical contribution, and empirical results suggest improved sensitivity to structural failures compared to traditional metrics. After rebuttal, the authors have adequately addressed concerns regarding robustness, computational efficiency, and experimental protocol for human perceptual alignment. The clarification on diagnostic capability further strengthens the paper.

However, several limitations remain. The main conceptual concern is that aggregating multiple aspects into a single scalar metric may obscure fine-grained evaluation signals, and the benefit over reporting separate metrics is not entirely conclusive. In addition, the work is limited to binary segmentation and is not directly applicable as a differentiable training objective, which constrains its broader impact.

Overall, the paper presents a solid and practically useful contribution to segmentation evaluation. While the novelty is moderate and the scope is somewhat constrained, the method is likely to be of interest to the community and could serve as a useful complementary evaluation tool.